# LEARNING TO COUNT WITHOUT ANNOTATIONS

## ABSTRACT

While recent supervised methods for reference-based object counting continue to improve the performance on benchmark datasets, they have to rely on small datasets due to the cost associated with manually annotating dozens of objects in images. We propose Unsupervised Counter (UnCo), a model that can learn this task without requiring any manual annotations. To this end, we construct "Self-Collages", images with various pasted objects as training samples, that provide a rich learning signal covering arbitrary object types and counts. Our method builds on existing unsupervised representations and segmentation techniques to successfully demonstrate for the first time the ability of reference-based counting without manual supervision. Our experiments show that our method not only outperforms simple baselines and generic models such as FasterRCNN and DETR, but also matches the performance of supervised counting models in some domains.

## 1 INTRODUCTION

Cognitive neuroscientists speculate that visual counting, especially for a small number of objects, is a pre-attentive and parallel process (Trick & Pylyshyn, 1994; Dehaene, 2011), which can help humans and animals make prompt decisions (Piazza & Dehaene, 2004). Accumulating evidence shows that infants and certain species of animals can differentiate between small numbers of items (Davis & Pérusse, 1988; Dehaene, 2011; Pahl et al., 2013) and as young as 18-month-old infants have been shown to develop counting abilities (Slaughter et al., 2011). These findings indicate that the ability of visual counting may emerge very early or even be inborn in humans and animals.

On the non-biological side, recent developments in computer vision have been tremendous. The state-of-the-art computer vision models can classify thousands of image classes (Krizhevsky et al., 2012; He et al., 2016), detect various objects (Zhou et al., 2022), or segment almost anything from an image (Kirillov et al., 2023). Partially inspired by how babies learn to see the world (Smith & Gasser, 2005), some of the recent well-performing models are trained with self-supervised learning methods, whereby a learning signal for neural networks is constructed without the need for manual annotations (Doersch et al., 2015; He et al., 2020). The pretrained visual representations from such methods have demonstrated superior performances on various downstream visual tasks, like image classification and object detection (He et al., 2020; Caron et al., 2021; He et al., 2022). Moreover, self-supervised learning signals have been shown to be sufficient for successfully learning image groupings (Yan et al., 2020; Van Gansbeke et al., 2020) and even object and semantic segmentations without any annotations (Caron et al., 2021; Zadaianchuk et al., 2023). Motivated by these, we ask in this paper whether visual counting might also be solvable without relying on human annotations.

The current state-of-the-art visual counting methods typically adapt pretrained visual representations to the counting task by using a considerable size of human annotations, *e.g.* CounTR from Liu et al. (2022). However, we conjecture that the existing visual representations are already *strong enough* to perform counting, even *without* any manual annotations.

In this paper, we design a straightforward self-supervised training scheme to teach the model 'how to count', by pasting a number of objects on a background image, to make a Self-Collage. Our experiments show that when constructing the Self-Collages carefully, this training method is effective enough to leverage the pretrained visual representation on the counting task, even approaching other methods that require manually annotated counting data. For the visual representation, we use the self-supervised pretrained DINO features (Caron et al., 2021), which have been shown to be useful and generalisable for a variety of visual tasks like segmenting objects (Melas-Kyriazi et al., 2022;

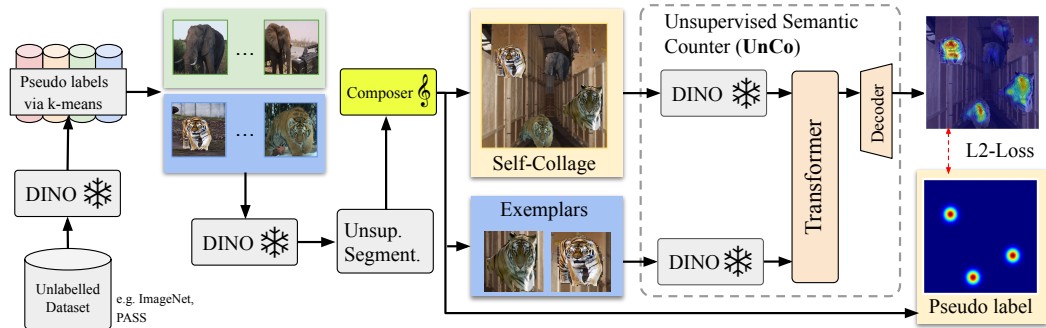

Figure 1: **Unsupervised Counter (UnCo) overview.** Our method leverages the strong coherence of deep clusters to provide pseudo-labelled images which are used to construct a self-supervised counting task. The composer utilises self-supervised segmentations for pasting a set of objects onto a background image and UnCo is trained to count these when provided with unsupervised exemplars.

Ziegler & Asano, 2022). Note that the DINO model is also trained without manual annotations, thus our entire pipeline does not require annotated datasets.

To summarise, this paper focuses on the objective of *training a semantic counting model without any manual annotation*. The following contributions are made: (i) We propose a simple yet effective data generation method to construct 'Self-Collages', which pastes objects onto an image and gets supervision signals for free. (ii) We leverage self-supervised pretrained visual features from DINO and develop UnCo, a transformer-based model architecture for counting. (iii) The experiments show that our method trained without manual annotations not only outperforms baselines and generic models like FasterRCNN and DETR, but also matches the performance of supervised counting models.

## 2 RELATED WORK

**Counting with object classes.** The class-specific counting methods are trained to count instances of a single class of interest, such as cars (Mundhenk et al., 2016; Hsieh et al., 2017) or people (Liu et al., 2018; Sam et al., 2020). These methods require retraining to apply them to new object classes (Mundhenk et al., 2016). In addition, some works rely on class-specific assumptions such as the distribution of objects which cannot be easily adapted (Sam et al., 2020).

By contrast, class-agnostic approaches are not designed with a specific object class in mind. Early work by Zhang et al. (2015) proposes the salient object subitizing task, where the model is trained to count by classification of $\{0, 1, 2, 3, 4+\}$ salient objects regardless of their classes. Other reference-less methods like Hobley & Prisacariu (2022) frame counting as a repetition-recognition task and aim to automatically identify the objects to be counted. An alternative approach of class-agnostic counting requires a prior of the object type to be counted in the form of reference images, also called 'exemplars', each containing a single instance of the desired class (Yang et al., 2021; Lu et al., 2018).

**Counting with different methods.** Categorised by the approach taken to obtain the final count, counting methods can be divided into classification, detection and regression-based methods.

Classification-based approaches predict a discrete count for a given image (Zhang et al., 2015). The classes either correspond to single numbers or potentially open-ended intervals. Thus, predictions are limited to the pre-defined counts and a generalisation to new ranges is by design not possible.

An alternative is detection-based methods (Hsieh et al., 2017). By predicting bounding boxes for the counted instances and deriving a global count based on their number, these methods are unlike classification-based approaches not constrained to predefined count classes. While the bounding boxes can facilitate further analysis by explicitly showing the counted instances, the performance of detection-based approaches deteriorates in high-density settings (Hobley & Prisacariu, 2022).

Lastly, regression-based methods predict a single number for each image and can be further divided into scalar and density-based approaches. Scalar methods directly map an input image to a single scalar count corresponding to the number of objects in the input (Hobley & Prisacariu, 2022).

Density-based methods on the contrary predict a density map for a given image and obtain the final count by integrating over it (Lu et al., 2018; Liu et al., 2018; Sam et al., 2020; Djukic et al., 2022; Chen et al., 2022). Similar to detection-based approaches, these methods allow locating the counted instances which correspond to the local maxima of the density map but have the added benefit of performing well in high-density applications with overlapping objects (Lu et al., 2018). The recent work CounTR (Liu et al., 2022) is a class-agnostic, density-based method which is trained to count both with and without exemplars using a transformer decoder structure with learnable special tokens.

**Self-supervised learning.** Self-supervised learning (SSL) has shown its effectiveness in many computer vision tasks. Essentially, SSL derives the supervision signal from the data itself rather than manual annotations. The supervision signal can be found from various "proxy tasks" like colorization (Zhang et al., 2016), spatial ordering or impaining (Pathak et al., 2016; Noroozi & Favaro, 2016), temporal ordering (Misra et al., 2016; Han et al., 2019), contrasting similar instances (Oord et al., 2018; Chen et al., 2020), clustering (Caron et al., 2018; Asano et al., 2020), and from multiple modalities (Radford et al., 2021; Alayrac et al., 2020). Another line of SSL methods is knowledge distillation, where one smaller student model is trained to predict the output of the other larger teacher model (Buciluǎ et al., 2006; Chen et al., 2017; Kim & Rush, 2016). BYOL (Grill et al., 2020) design two identical models but train one model to predict the moving average of the other as the supervision signal. Notably, DINO (Caron et al., 2021) is trained in a similar way but using Vision Transformers (ViTs) as the visual model (Dosovitskiy et al., 2021), and obtains strong visual features. Simply thresholding the attention maps and using the resulting masks obtains superior-quality image segmentations. Follow-up works (Shin et al., 2022; Ziegler & Asano, 2022) demonstrate the semantic segmentation quality can be further improved by applying some light-weight training or post-processing of DINO features. For counting in SSL, Noroozi et al. (2017) use feature "counting" as a proxy task to learn representations for transfer learning. In this work, we focus on counting itself and explore approaches to teach the model to count without manual supervision.

## 3 METHOD

We tackle the task of counting objects given some exemplar crops within an image. With an image dataset $\mathcal{D} = \{(\boldsymbol{I}_1, \mathcal{S}_1, \mathbf{y}_1), \dots, (\boldsymbol{I}_i, \mathcal{S}_i, \mathbf{y}_i)\}$, where $\boldsymbol{I}_i \in \mathbb{R}^{H \times W \times 3}$ denotes image $i$, $\mathcal{S}_i = \{\boldsymbol{E}_i{}^1, \dots, \boldsymbol{E}_i{}^E\}$ represent the $E$ visual exemplars of shape $\boldsymbol{E}_i{}^e \in \mathbb{R}^{H' \times W' \times 3}$, and $\mathbf{y}_i \in \mathbb{R}^{H \times W}$ is the ground-truth density map, the counting task can be written as:

$$\hat{\mathbf{y}}_i = f_\Theta(\boldsymbol{I}_i, \mathcal{S}_i) \tag{1}$$

Here, the predicted density map $\hat{\mathbf{y}}_i$ indicates the objects to be counted, as specified by the exemplars $\mathcal{S}_i$, such that $\sum_{kl} \hat{\mathbf{y}}_{i_{kl}}$ yields the overall count for the image $\boldsymbol{I}_i$. We are interested in training a neural network $f$ parameterised by $\Theta$ to learn how to count based on the exemplars $\mathcal{S}_i$. For the supervised counting methods (Liu et al., 2022; Lu et al., 2018), the network parameters $\Theta$ can be trained with the '(`prediction`, `ground-truth`)' pairs: $(\hat{\mathbf{y}}_i, \mathbf{y}_i)$. However, for self-supervised counting, the learning signal $\mathbf{y}_i$ is not obtained from manual annotations, but instead from the data itself.

In this section, we introduce two essential parts of our method: we start by presenting our data generation method for counting in Section 3.1, that is the construction of the tuple $(\boldsymbol{I}_i, \mathcal{S}_i, \mathbf{y}_i)$; then, we explain the Unsupervised Counter (UnCo) model, *i.e.* $f_\Theta$ in Equation (1), in Section 3.2. An overview of our method is provided in Figure 1.

### 3.1 CONSTRUCTING SELF-COLLAGES

A key component of self-supervised training is the construction of a supervision signal without any manual annotations. In this paper, we generate training samples by pasting different images on top of a background. Unlike other works which combine annotated training images to enrich the training set (Hobley & Prisacariu, 2022; Liu et al., 2022), we use this idea to construct the *whole* training set including unsupervised proxy labels, yielding self-supervised collages, or Self-Collages for short. The generation process is described by a composer module $g$, yielding a distribution $g(\mathcal{O}, \mathcal{B}) = p(\tilde{\boldsymbol{I}}, \mathcal{S}, \mathbf{y} \mid \mathcal{O}, \mathcal{B})$ of constructed images $\tilde{\boldsymbol{I}} \in \mathbb{R}^{H \times W \times 3}$ along with unsupervised exemplars $\mathcal{S}$ and labels $\mathbf{y}$ based on two sets of unlabelled images $\mathcal{O}$ and $\mathcal{B}$. $\mathcal{S} = \{\boldsymbol{E}\}^E$, $\boldsymbol{E} \in \mathbb{R}^{H' \times W' \times 3}$ is a set of $E \in \mathbb{N}$ exemplars and $\mathbf{y} \in \mathbb{R}^{H \times W}$ corresponds to the density map of $\tilde{\boldsymbol{I}}$.

The composer module $g$ first randomly selects the number of distinct object categories $n_c \sim U[t_{\min}, t_{\max}]$ the first of which is taken as the target cluster. To reduce the risk of overfitting to construction artefacts, we always construct images with $n_{\max}$ objects and change the associated number of target objects $\sum_{ij} \mathbf{y}_{ij}$ solely by altering the number of objects in the target cluster. This way, the number of pasted objects and, therefore, the number of artefacts is independent of the target count. The number of target objects $n_0 = n \sim U[n_{\min}, n_{\max} - n_c + 1]$ has an upper-bound lower than $n_{\max}$ to guarantee that there is at least one object of each of the $n_c$ types. For all other clusters, the number of objects is drawn from a uniform distribution of points on the $n_c - 1$ dimensional polytope with $L1$-size of $n_{\max} - n$ ensuring that the total number of objects equals $n_{\max}$. Further details, including the pseudocode for the composer module are shown in Appendix A.2.

**Unsupervised categories.** We obtain self-supervised object categories by first extracting feature representations for all samples in $\mathcal{O}$ using a pretrained DINO ViT-B/16 backbone (Caron et al., 2021) $d$ and subsequently running k-means with $K$ clusters:

$$c(\boldsymbol{I}) = \text{k-means}_K[(d(\mathcal{O}))^{\text{CLS}}](\boldsymbol{I}), \tag{2}$$

where $c(\boldsymbol{I})$ is the unsupervised category for image $\boldsymbol{I}$ constructed using the final $\text{CLS}$-embedding of $d$. For each of the $n_c$ clusters, a random, unique cluster $c_i$, $i \in [0, n_c - 1]$ is chosen from all $K$ clusters where $c_0$ is the target cluster.

**Image selection.** In the next step, random sets of images $\mathcal{I}_i \subset \mathcal{O}$ are picked from their corresponding unsupervised categories $c_i$, such that $|\mathcal{I}_i| = n_i$ and $c(\boldsymbol{I}) = c_i \; \forall \boldsymbol{I} \in \mathcal{I}_i$. We denote the union of these sets as $\mathcal{I} = \bigcup_{i=0}^{n_c - 1} \mathcal{I}_i$. In addition, we sample one image $\boldsymbol{I}_b$ from another dataset $\mathcal{B}$, which is assumed to not contain salient objects to serve as the background image.

### 3.1.1 CONSTRUCTION STRATEGY

Here we detail the Self-Collage construction. First, the background image $\boldsymbol{I}_b$ is reshaped to the final dimensions $H \times W$ and used as a canvas on which the modified images $\mathcal{I}$ are pasted. To mimic natural images that typically contain similarly sized objects, we first randomly pick a mean object size $d_{\text{mean}} \sim U[d_{\min}, d_{\max}]$. Subsequently, the target size of the pasted objects is drawn independently for each $\boldsymbol{I} \in \mathcal{I}, \boldsymbol{I} \in \mathbb{R}^{d_h \times d_w \times 3}$ from a uniform distribution $d_{\text{paste}} \sim U[(1 - \sigma) \cdot d_{\text{mean}}, (1 + \sigma) \cdot d_{\text{mean}}]$ where $\sigma \in (0, 1)$ controls the diversity of objects sizes in an image. After resizing $\boldsymbol{I}$ to $\boldsymbol{I}_r \in \mathbb{R}^{d_{\text{paste}} \times d_{\text{paste}} \times 3}$, the image is pasted to a random location on $\boldsymbol{I}_b$. This location is either a position where previously no object has been pasted, or any location in the constructed image, potentially leading to overlapping images. By default, we will use the latter.

**Segmented pasting.** Since pasting the whole image $\boldsymbol{I}_r$ might violate the assumption of having a single object and results in artefacts by also pasting the background of $\boldsymbol{I}_r$, we introduce an alternative construction method using self-supervised segmentations. This method uses an unsupervised segmentation method (Shin et al., 2022) to obtain a noisy foreground segmentation $\mathbf{s} \in [0, 1]^{d_h \times d_w}$ for $\boldsymbol{I}$. Instead of pasting the whole image, the segmentation $\mathbf{s}$ is used to only copy its foreground. Additionally, having access to the segmentation $\mathbf{s}$, this method can directly control the size of the pasted objects rather than the pasted images. To do that, we first extract the object in $\boldsymbol{I}$ by computing the Hadamard product $\boldsymbol{I}_{\text{object}} = \text{cut}(\boldsymbol{I} \circ \mathbf{s})$ where the operation "cut" removes rows and columns that are completely masked out. In the next step, $\boldsymbol{I}_{\text{object}} \in \mathbb{R}^{h_{\text{object}} \times w_{\text{object}} \times 3}$ is resized so that the maximum dimension of the resized object equals $d_{\text{paste}}$ before pasting it onto $\boldsymbol{I}_b$. We use this setting by default.

**Exemplar selection.** To construct the exemplars used for training, we exploit the information about how the sample was constructed using $g$: The set of $E$ exemplars $\mathcal{S}$ is simply obtained by filtering for pasted objects that belong to the target cluster $c_0$ and subsequently sampling $E$ randomly. Then, for each of them, a crop of $\tilde{\boldsymbol{I}}$ is taken as exemplar after resizing its spatial dimensions to $H' \times W'$.

### 3.1.2 DENSITY MAP CONSTRUCTION

To train our counting model, we construct an unsupervised density map $\mathbf{y}$ as a target for each training image $\tilde{\boldsymbol{I}}$. This density map needs to have the following two properties: i) it needs to sum up to the overall count of objects that we are counting and ii) it needs to have high values at object locations. To this end, we create $\mathbf{y}$ as a simple density map of Gaussian blobs as done in supervised density-based counting methods (Djukic et al., 2022; Liu et al., 2022). For this, we use the bounding box for each pasted target image $\boldsymbol{I} \in \mathcal{I}_0$ and place Gaussian density at the centre and normalise it to one.

## 3.2 Unsupervised Counter

**Model architecture.** UnCo's architecture is inspired by CounTR (Liu et al., 2022). To map an input image $I$ and its exemplars $\mathcal{S}$ to a density map $\hat{\mathbf{y}}$, the model consists of four modules: image encoder $\Phi$, exemplar encoder $\Psi$, feature interaction module $f_{\text{fim}}$, and decoder $f_{\text{dec}}$. An overview of this architecture can be seen in Figure 1. The image encoder $\Phi$ encodes an image $I$ into a feature map $\mathbf{x} = \Phi(I) \in \mathbb{R}^{h \times w \times d}$ where $h, w, d$ denote the height, width, and channel depth. Similarly, each exemplar $E \in \mathcal{S}$ is projected to a single feature vector $\mathbf{z} \in \mathbb{R}^d$ by taking a weighted average of the grid features. Instead of training a CNN for an exemplar encoder as CounTR does, we choose $\Psi = \Phi$ to be the frozen DINO visual encoder weights. Reusing these weights, the exemplar and image features are in the same feature space. The feature interaction module (FIM) $f_{\text{fim}}$ enriches the feature map $\mathbf{x}$ with information from the encoded exemplars $\mathbf{z}_j$, $j \in \{1, ..., E\}$ with a transformer decoder structure. Finally, the decoder $f_{\text{dec}}$ takes the resulting patch-level feature map of the FIM as input and upsamples it with 4 convolutional blocks, ending up with a density map of the same resolution as the input image. Please refer to Appendix A.1 for the full architectural details.

**UnCo supervision.** UnCo is trained using the Mean Squared Error (MSE) between model prediction and pseudo ground-truth. Given a Self-Collage $\tilde{I}$, exemplars $\mathcal{S}$, and density map $\mathbf{y}$, the loss $\mathcal{L}$ for an individual sample is computed using the following equation where $f_{\Theta}(\tilde{I}, \mathcal{S})_{ij}$ is UnCo's spatially-dense output at location $(i, j)$:

$$\mathcal{L} = \frac{1}{H * W} \sum_{ij} (\mathbf{y}_{ij} - f_{\Theta}(\tilde{I}, \mathcal{S})_{ij})^2 \tag{3}$$

## 4 Experiments

### 4.1 Implementation Details

**Datasets.** To construct Self-Collages, we use **ImageNet-1k** (Deng et al., 2009) and **SUN397** (Xiao et al., 2010). ImageNet-1k contains 1.2M mostly object-centric images spanning 1000 object categories. SUN397 contains 109K images for 397 scene categories like 'cliff' or 'corridor'. Note that the object or scene category information is never used in our method. We assume that images from ImageNet-1k contain a single salient object and images from SUN397 contain no salient objects to serve as sets $\mathcal{O}$ and $\mathcal{B}$. Based on this, $g$ randomly selects images from SUN397 as background images and picks objects from ImageNet-1k. Although Imagenet-1k and SUN397 contain on average 3 and 17 objects respectively (Lin et al., 2014), these assumptions are still reasonable for our data construction. Appendix B.1 shows examples of both datasets and discusses these assumptions.

To evaluate the counting ability, we use the standard **FSC-147** dataset (Ranjan et al., 2021), which contains 6135 images covering 147 object categories, with counts ranging from 7 to 3731 and an average count of 56 objects per image. For each image, the dataset provides at least three randomly chosen object instances with annotated bounding boxes as exemplars. To analyse the counting ability in detailed count ranges, we partition the FSC-147 test set into three parts of roughly 400 images each, resulting in **FSC-147-{low,medium,high}** subsets, each covering object counts from 8-16, 17-40 and 41-3701 respectively. Unless otherwise stated, we evaluate using 3 exemplars.

Additionally, we also use the **CARPK** (Hsieh et al., 2017) and the Multi-Salient-Object (**MSO**) dataset (Zhang et al., 2015) for evaluation. CARPK consists of 459 images of parking lots. Each image contains between 2 and 188 cars, with an average of 103. MSO contains 1224 images covering 5 categories: {0,1,2,3,4+} salient objects, with the bounding boxes of salient objects annotated. This dataset is largely imbalanced as 338 images contain zero salient objects and 20 images contain at least 4. For evaluation, we removed all samples with 0 counts, split the 4+ category into exact counts based on the number of annotated objects, and chose only one annotated object as exemplar.

**Construction details.** We configure the composer module to construct training samples using objects of $K = 10,000$ different clusters. To always have objects of a target and a non-target cluster in each image, we set $t_{\min} = t_{\max} = 2$. Since the minimum number of target objects in an image limits the maximum number of exemplars available during training, we set $n_{\min} = 3$, the maximum is $n_{\max} = 20$. Finally, we choose $d_{\min} = 15$, $d_{\max} = 70$, and $\sigma = 0.3$ to obtain diverse training images with objects of different sizes. The code is provided as part of the Supplementary Material.

Table 1: **Comparison to baselines.** Evaluation on different FSC-147 test subsets. "Conn. Comp." refers to connected components on DINO's attention map.

| Method | FSC-147 **low** | | | FSC-147 **medium** | | | FSC-147 **high** | | |
|---|---|---|---|---|---|---|---|---|---|
| | MAE↓ | RMSE↓ | $\tau\uparrow$ | MAE↓ | RMSE↓ | $\tau\uparrow$ | MAE↓ | RMSE↓ | $\tau\uparrow$ |
| Average | 37.71 | 37.79 | - | 22.74 | 23.69 | - | 68.88 | 213.08 | - |
| Conn. Comp. | 14.71 | 18.59 | 0.14 | 14.19 | 17.90 | 0.16 | 69.77 | 210.54 | 0.17 |
| FasterRCNN | 7.06 | 8.46 | -0.03 | 19.87 | 22.25 | -0.04 | 109.12 | 230.35 | -0.06 |
| DETR | 6.92 | **8.20** | 0.07 | 19.33 | 21.56 | -0.08 | 109.34 | **162.88** | -0.07 |
| **UnCo (ours)** | **5.60** | 10.13 | **0.27** | **9.48** | **12.73** | **0.34** | **67.17** | 189.76 | **0.26** |
| $\sigma$(5 runs) | ±0.48 | ±0.84 | ±0.02 | ±0.19 | ±0.33 | ±0.02 | ±1.03 | ±1.38 | ±0.01 |

**Training and inference details.** During training, the images are cropped and resized to $224 \times 224$ pixels. The exemplars are resized to $64 \times 64$ pixels. For each batch, we randomly draw the number of exemplars $E$ to be between 1 and 3. We follow previous work (Liu et al., 2022) by scaling the loss, in our case by a factor of 3,000, and randomly dropping 20% of the non-object pixels in the density map to decrease the imbalance of object and non-object pixels. By default, we use an AdamW optimizer and a cosine-decay learning rate schedule with linear warmup and a maximum learning rate of $5 \times 10^{-4}$. We use a batch size of 128 images. Each model is trained on an Nvidia A100 GPU for 50 epochs of $10,000$ Self-Collages each, which takes about 4 hours. At inference, we use a sliding window approach similar to Liu et al. (2022), see Appendix A.4 for details. For evaluation metrics, we report Mean Absolute Error (MAE) and Root Mean Squared Error (RMSE) following previous work (Liu et al., 2022; Djukic et al., 2022). We also report Kendall's $\tau$ coefficient, which is a rank correlation coefficient between the sorted ground-truth counts and the predicted counts.

**Baselines.** To verify the effectiveness of our method, we introduce a series of baselines to compare with. (1) **Average** baseline: the prediction is always the average count of the FSC-147 training set, which is 49.96 objects. (2) **Connected components** baseline: for this, we use the final-layer attention values from the CLS-token of a pretrained DINO ViT-B/8 model. To derive a final count, we first threshold the attention map of each head to keep $p_{att}$ percent of the attention mask. Subsequently, we consider each patch to where at least $n_{head}$ attention heads attend to belong to an object. The number of connected components in the resulting feature map which cover more than $p_{size}$ percent of the feature map is taken as prediction. To this end, we perform a grid search with almost 800 configurations of different value combinations for the three thresholds $p_{att}$, $n_{head}$, and $p_{size}$ on the FSC-147 training set and select the best configuration based on the MAE. The specific values tested can be found in Appendix A.3. (3) **FasterRCNN** baseline: we run the strong image detection toolbox FasterRCNN (Ren et al., 2015) with a score threshold of 0.5 on the image, which predicts a number of object bounding boxes. Then the object count is obtained by parsing the detection results by taking the total number of detected bounding boxes. Just like the connected components baseline, this model is applied to images resized to $384 \times 384$ pixels. (4) **DETR** baseline: similar to FasterRCNN, we evaluate the detection model DETR (Carion et al., 2020) on the counting task. Here, we resize the images to $800 \times 800$ pixels to better match DETR's evaluation protocol.

## 4.2 COMPARISON AGAINST BASELINES

In Table 1, we compare against multiple baselines on the three equal-sized splits of FSC-147. As the connected components cannot leverage the additional information of the few-shot samples provided, we try to make the comparison as fair as possible, by testing almost 800 threshold parameters on the FSC-147 training set. While this yields a strong baseline, we find that our method of learning with Self-Collages more than halves the MAE on the *low* split, despite using a similar visual backbone. Next, we compare against the FasterRCNN and DETR object detectors which, unlike UnCo, are trained in a supervised fashion. Even though DETR outperforms UnCo in terms of RMSE on FSC-147 low and high, we find that our method still outperforms DETR as well as all other baselines on 7 out of 9 measures. This is despite the additional advantages such as access to the FSC-147 training distribution for these baselines. The gap between most baselines and UnCo is the smallest on the *high* split, suggesting limits to its generalisation which we will explore further in Appendix C.1. We also provide a qualitative analysis of some failure cases of FasterRCNN in Appendix C.4.

Table 2: **Ablations.** We ablate various components of our model architecture and Self-Collage construction method. Default settings are highlighted in grey.

| Frozen $\Phi$ | Frozen $\Psi$ | FSC-147 MAE↓ | RMSE↓ | FSC-147 **low** MAE↓ | RMSE↓ |
|:---:|:---:|:---:|:---:|:---:|:---:|
| ✗ | ✗ | 37.64 | **126.91** | 7.99 | 13.37 |
| ✓ | ✗ | 36.94 | 130.22 | 9.10 | 14.69 |
| ✓ | ✓ | **35.77** | 130.34 | **5.60** | **10.13** |

(a) Keeping both image encoder $\Phi$ and exemplar encoder $\Psi$ frozen works best.

| Segm. | Overlap. | FSC-147 MAE↓ | RMSE↓ | FSC-147 **low** MAE↓ | RMSE↓ |
|:---:|:---:|:---:|:---:|:---:|:---:|
| ✓ | ✗ | 36.47 | **128.97** | 8.84 | 14.54 |
| ✗ | ✓ | 37.46 | 136.03 | 6.33 | 11.55 |
| ✓ | ✓ | **35.77** | 130.34 | **5.60** | **10.13** |

(b) Using both segmented objects and allowing for overlapping objects works well.

| $n_{max}$ | FSC-147 MAE↓ | RMSE↓ | FSC-147 **low** MAE↓ | RMSE↓ |
|:---:|:---:|:---:|:---:|:---:|
| 50 | **30.17** | **115.42** | 6.65 | 13.60 |
| 20 | 35.77 | 130.34 | **5.60** | **10.13** |

(c) The number of pasted objects correlates with the final model performance.

| #Exemplars | FSC-147 MAE↓ | RMSE↓ | FSC-147 **low** MAE↓ | RMSE↓ |
|:---:|:---:|:---:|:---:|:---:|
| 0 - 3 | 36.28 | 131.44 | 6.23 | 10.33 |
| 1 - 3 | **35.77** | **130.34** | **5.60** | **10.13** |

(d) Training our method strictly in the 1-3 exemplar setting works best.

Table 3: **Combining Self-Collages with different backbones.** We can apply our method to any recent state-of-the-art pretrained network and achieve strong performances.

| Architecture | Pretraining | FSC-147 MAE↓ | RMSE↓ | $\tau$ ↑ | FSC-147 **low** MAE↓ | RMSE↓ | $\tau$ ↑ |
|:---|:---|:---:|:---:|:---:|:---:|:---:|:---:|
| ViT-B/8 | Leopart (Ziegler & Asano, 2022) | 36.04 | 126.75 | 0.55 | 5.72 | 10.67 | 0.25 |
| ViT-B/8 | DINO (Caron et al., 2021) | 38.16 | 132.09 | 0.50 | 6.10 | 9.79 | 0.22 |
| ViT-B/16 | DINO (Caron et al., 2021) | 35.77 | 130.34 | 0.57 | 5.60 | 10.13 | 0.27 |

## 4.3 ABLATION STUDY

We analyse the various components of our method in Tables 2a to 2d.

**Keeping backbone frozen works best.** In Table 2a, we evaluate the effect of unfreezing the last two backbone blocks shared by $\Phi$ and $\Psi$. In addition, we test training a CNN-based encoder $\Psi$ from scratch similar to Liu et al. (2022). We find that keeping both frozen, *i.e.* $\Phi = \Psi = \text{const.}$ works best, as it discourages the visual features to adapt to potential artefacts in our Self-Collages.

**Benefit of self-supervised exemplars.** In Table 2d we evaluate the effect of including the zero-shot counting task (*i.e.* counting all salient objects) as done in other works (Liu et al., 2022; Djukic et al., 2022). However, we find that including this task even with only a 25% chance leads to a lower performance. This is likely due to the zero-shot task overfitting our Self-Collages and finding short-cuts, such as pasting artefacts, as we relax the constraint introduced in Section 3.1.1 and vary the number of pasted objects to match the desired target. We therefore train our model with 1-3 exemplars and show how we can still conduct semantic zero-shot counting in Section 4.6.

**Maximum number of pasted objects.** Next, we evaluate the effect of varying $n_{max}$, the maximum number of objects pasted. Pasting up to 50 objects yields overall better performance on the full FSC-147 test dataset which contains on average 66 objects. While this shows that the construction of Self-Collages can be successfully scaled to higher counts, we find that pasting with 20 objects already achieves good performance with shortened construction times and so use this setting.

**Segmented pasting works best.** From Table 2b, we find that the best construction strategy involves pasting the self-supervised segmentations, regardless of their overlap with other objects. We simply store a set of segmentations and combine these to create diverse Self-Collages at training time.

**Compatible with various pretrained backbones.** In Table 3, we show the effect of using different frozen weights for the visual encoder. Across ViT-B models with varying patch sizes and pretrainings, from Leopart's (Ziegler & Asano, 2022) spatially dense to the original DINO's (Caron et al., 2021) image-level pretraining, we find similar performances across our evaluations. This shows that our method is compatible with various architectures and pretrainings and will likely benefit from the

Table 4: **Comparison to supervised models.** We evaluate on the val and test split of FSC-147.

| Methods | Val | | Test | |
|---|---|---|---|---|
| | MAE↓ | RMSE↓ | MAE↓ | RMSE↓ |
| CounTR (Liu et al., 2022) | 13.13 | 49.83 | 11.95 | 91.23 |
| LOCA (Djukic et al., 2022) | **10.24** | **32.56** | **10.79** | **56.97** |
| **UnCo (ours)** | $36.93_{\pm0.49}$ | $106.61_{\pm1.46}$ | $35.77_{\pm0.60}$ | $130.34_{\pm0.94}$ |

Table 5: **Comparison to CounTR.** Evaluation results on the FSC-147 *low* and *medium* test subsets.

| Method | FSC-147 **low** (8-16 objects) | | | | | FSC-147 **medium** (17-40 objects) | | | | |
|---|---|---|---|---|---|---|---|---|---|---|
| | MAE↓ | Δ | RMSE↓ | Δ | τ ↑ | MAE↓ | Δ | RMSE↓ | Δ | τ ↑ |
| CounTR | 6.58 | | 48.11 | | **0.42** | **4.48** | | **10.02** | | **0.60** |
| **UnCo (ours)** | $\mathbf{5.60}_{\pm0.48}$ | -0.98 | $\mathbf{10.13}_{\pm0.84}$ | -37.9 | $0.27_{\pm0.02}$ | $9.48_{\pm0.19}$ | +5.0 | $12.73_{\pm0.33}$ | +2.7 | $0.34_{\pm0.02}$ |

continued progress in self-supervised representation learning which is supported by our findings in Appendix C.2. We do not find any benefit in going from patch size 16 to 8 (similar to Siméoni et al. (2021)) so we use the faster DINO ViT-B/16 model for the rest of the paper.

### 4.4 BENCHMARK COMPARISON

Next, we compare with previous methods on three datasets: FSC-147 (Ranjan et al., 2021), MSO (Zhang et al., 2015), and CARPK (Hsieh et al., 2017). Table 4 shows the results on the validation and test split of the FSC-147 dataset. Note that the comparison is certainly not fair because of two factors: (1) our method does not use manually annotated counting data, also has never seen any FSC-147 training images, (2) our training samples based on Self-Collages only cover 3-19 target objects, whereas the full test set of FSC-147 has 66 objects on average. Evaluating our method on the full set of FSC-147 is actually evaluating the transferring ability as discussed in Appendix C.1.

We choose CounTR (Liu et al., 2022) for further analysis because of its similar architecture to UnCo. To fairly compare with this method, we trisect the FSC-147 test set as described in Section 4.1. Table 5 shows the evaluation results on the *low* and *medium* partitions. It is remarkable that our method outperforms supervised CounTR on the *low* partition, especially on the RMSE metric (10.13 vs 48.11). Also, our method is not far from CounTR's performance on the *medium* partition (*e.g.* RMSE 12.7 vs 10.02), showing a reasonable transferring ability since the model is trained with up to 19 object counts only.

Additionally, Table 6 compares our method with CounTR on a subset of MSO dataset as described in Section 4.1. The results show our self-supervised method outperforms the supervised CounTR model by a large margin on the lower counts tasks. In Table 7, the performance on the CARPK dataset is shown. We use the same CounTR model finetuned on the FSC-147 dataset as before to com-

Table 6: **Evaluation on the MSO Zhang et al. (2015) dataset.**

| Method | MSO | | |
|---|---|---|---|
| | MAE↓ | RMSE↓ | τ ↑ |
| CounTR | 2.34 | 8.12 | 0.36 |
| **UnCo (ours)** | **1.07** | **2.32** | **0.45** |
| σ(5 runs) | ±0.06 | ±0.25 | ±0.01 |

Table 7: **Evaluation on the CARPK Hsieh et al. (2017) dataset.**

| Method | CARPK | | |
|---|---|---|---|
| | MAE↓ | RMSE↓ | τ ↑ |
| CounTR | 19.62 | 29.70 | 0.57 |
| **UnCo (ours)** | 30.35 | 35.67 | 0.54 |
| σ(5 runs) | ±2.26 | ±2.36 | ±0.06 |

pare the generalisation across datasets. While the MAE of CounTR almost doubles compared to the results on the FSC-147 test set, UnCo's performance is more stable, improving the MAE by 15%.

### 4.5 QUALITATIVE RESULTS AND LIMITATIONS

Figure 2 shows a few qualitative results to demonstrate our method's effectiveness and its limitations. The model is able to correctly predict the number of objects for clearly separated and slightly overlapping instances (*e.g.* Subfigures **a** and **c**). The model also successfully identifies the object type of interest, *e.g.* in Subfigure **b** the density map correctly highlights the strawberries rather than blueberries. One limitation of our model is partial or occluded objects. For example, in Subfig-

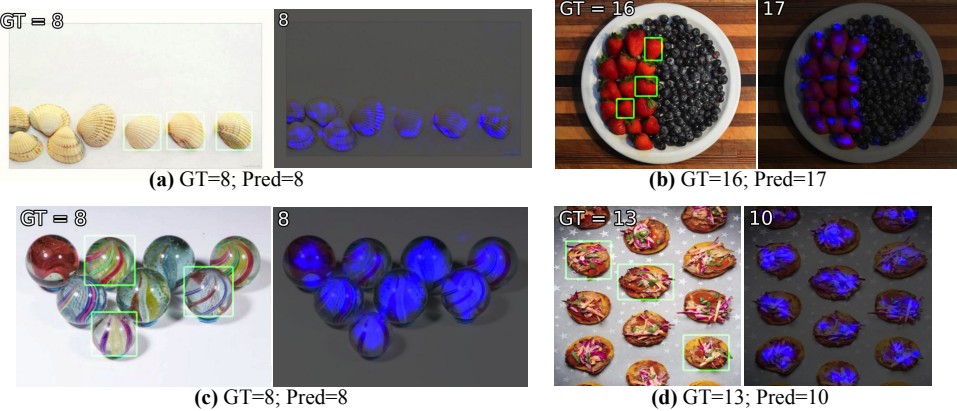

**(a)** GT=8; Pred=8      **(b)** GT=16; Pred=17

**(c)** GT=8; Pred=8      **(d)** GT=13; Pred=10

Figure 2: **Qualitative examples of UnCo predictions.** We show predictions on four images from the FSC-147 test set, the green boxes represent the exemplars. Our predicted count is the sum of the density map rounded to the nearest integer.

ure **d** the prediction missed a few burgers which are possibly the ones partially shown on the edge. However, partial or occluded objects are also challenging and ambiguous for humans.

### 4.6 SELF-SUPERVISED SEMANTIC COUNTING

In this last section, we explore the potential of UnCo for more advanced counting tasks. In particular, we test whether our model can not only unsupervisedly count different kinds of objects in an image, but also determine the categories by itself, a scenario we refer to as *semantic* counting. To this end, we use a simple pipeline that picks an area surrounding the maximum in the input image's `CLS`-attention map as the first input and refines it to obtain an exemplar. Next, UnCo predicts the number of objects in the image based on the self-constructed exemplar. Finally, the locations that have been detected using this procedure are removed from the initial attention map and if the remaining attention values are high enough, the process is repeated to count objects of another type. We provide the details of this method in Appendix A.5. In Figure 3, we demonstrate results on two difficult real images.

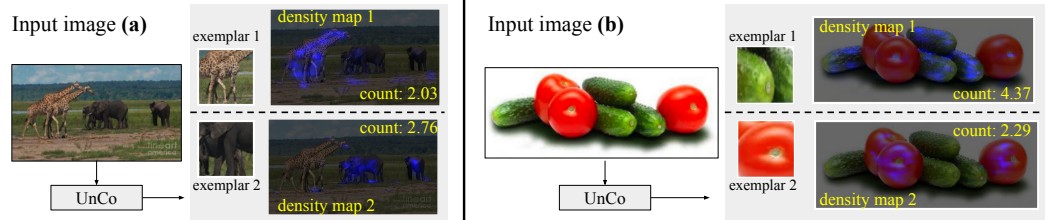

Figure 3: **Self-supervised semantic counting.** In this setting, the model proposes the exemplars by itself and then performs reference-based counting.

## 5 CONCLUSION

In this work, we have introduced Unsupervised Counter (UnCo), which is trained to count objects without any human supervision. To this end, our method constructs Self-Collages, a simple unsupervised way of creating proxy learning signals from unlabeled data. Our results demonstrate that by utilising an off-the-shelf unsupervisedly pretrained visual encoder, we can learn counting models that can even outperform strong baselines such as DETR and achieve similar performances to dedicated counting models such as CounTR on CARPK, MSO, and various splits of FSC-147. Finally, we have shown that our model can unsupervisedly identify multiple exemplars in an image and count them, something no supervised model can yet do. This work opens a wide space for extending unsupervised visual understanding beyond simple image-level representations to more complex tasks previously out of reach, such as scene graphs or unsupervised semantic instance segmentations.

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

# A   FURTHER IMPLEMENTATION DETAILS

## A.1   MODEL ARCHITECTURE

**Image encoder**   For the image encoder $\Phi$, we employ a ViT-B/16 pretrained using the DINO approach (Caron et al., 2021) as the backbone. It consists of 12 transformer blocks with 12 heads each and uses fixed, sinusoidal position embeddings. The transformer operates on $d = 768$ dimensions and increases the hidden dimensions within the two-layer MLPs after each attention block by a factor of four to 3072 dimensions. Each linear layer in the MLP is followed by the GELU non-linearity. More information can be found in the original work from Caron et al. (2021). By default, we freeze all 86M parameters of the backbone.

**Exemplar encoding**   To encode an exemplar $E$ with fixed spatial dimensions $H' \times W'$ into a single feature vector $\mathbf{z} \in \mathbb{R}^d$, we first pass it through the backbone of the exemplar encoder $\Psi$ to obtain a feature map $\mathbf{x}_E = \Psi(E) \in \mathbb{R}^{h \times w \times d}$ where $\Psi = \Phi$. The final representation $\mathbf{z}$ is derived by computing the weighted sum of $\mathbf{x}_E$ across the spatial dimensions where the weight of each patch is determined by the attention in the final `CLS`-attention map of $\Psi$ averaged over the heads.

**Feature interaction module**   We follow the architecture proposed by Liu et al. (2022) which uses 2 transformer blocks with 16 heads to modify the image embeddings by using self-attention. In addition to self-attention, each block utilises cross-attention where the keys and values are based on the encoded exemplars. Since this transformer operates on 512-dimensional feature vectors, $\mathbf{x}$ and $\mathbf{z}_j$, $j \in \{1, ..., E\}$ are projected to these dimensions using a linear layer. To give the feature interaction module direct access to positional information, fixed, sinusoidal position embeddings are added to the feature map. Similar to the image encoder, the MLPs increase the dimensionality four times to 2048. This results in 8.8M parameters.

**Decoder**   The decoder is built based on 4 convolutional layers that upscale the patch-level features to the original resolution to obtain the final density map. It has a total of 3.0M parameters. All blocks contain a convolutional layer with 256 output channels and a kernel size of $3 \times 3$. Each of them is followed by group normalisation with 8 groups and a ReLU non-linearity. The last block has a final convolutional layer with $1 \times 1$ filters that reduce the number of channels to 1 to match the desired output format. After each block, the spatial resolution is doubled using bilinear interpolation. The result is a density map $\hat{\mathbf{y}} \in \mathbb{R}^{H \times W}$ with the same resolution as the input image.

## A.2   COMPOSER MODULE DETAILS

We always place the images $\mathcal{I}_0$ of the target cluster $c_0$ on top of the non-target images $\mathcal{I}_i$, $i \in \{1, ..., n_c - 1\}$. This reduces the noise of Self-Collages since target objects can only be occluded by other target objects but never completely hidden by non-target images. Crucially, this also guarantees that exemplars are never covered by non-target objects which could alter the desired target cluster. While the resulting Self-Collages exhibit pasting artefacts (see Appendix B.2), previous work has shown that eliminating these artefacts by applying a blending method does not improve the performance on downstream tasks such as object detection (Dvornik et al., 2018) and instance segmentation (Ghiasi et al., 2021; Zhao et al., 2022). Hence, we focus on simple pasting to keep the complexity of the construction process low and rely on breaking correlations between the number of artefacts and the target count of an image by pasting a constant number of images as discussed in Section 3.1. Algorithm 1 shows the pseudocode for the composer module.

If we use the "no-overlap" setup, which prevents objects are pasted on top of each other, and an image cannot be pasted without overlapping with an already copied image, the construction process is restarted with new random sizes and locations. To guarantee the termination of $g$, potential overlaps between images are ignored if the construction process fails 20 times.

**Density map construction**   We construct the density map by placing unit density at the centre of each pasted target image $I \in \mathcal{I}_0$. Following Djukic et al. (2022), we apply a Gaussian filter to blur the resulting map. Its kernel size and standard deviation vary per image and are based on the average bounding box size divided by 8.

---

**Algorithm 1** The composer module

---

**Require:** $t_{\min}$       ▷ minimum number of clusters
         $t_{\max}$       ▷ maximum number of clusters
         $n_{\min}$       ▷ minimum number of target objects
         $n_{\max}$       ▷ maximum number of objects
         $K$       ▷ total number of clusters
         $E$       ▷ number of exemplars
         $\mathcal{O}$       ▷ set of object images
         $\mathcal{B}$       ▷ set of background images

---

**# create clusters and select cluster sizes**
$\mathcal{C} \leftarrow \text{k-means}_K[(d(\mathcal{O}))^{\text{CLS}}]$       ▷ cluster $\mathcal{O}$ using $K$ clusters
$n_c \sim U[t_{\min}, t_{\max}]$       ▷ select the number of clusters
$n_0 = n \sim U[n_{\min}, n_{\max} - n_c + 1]$       ▷ select the number of objects in the target cluster
**for** $i$ in range$(1, n_c - 1)$ **do**       ▷ select the number of objects in the other clusters

$$n_i \sim U\left[1, n_{\max} - \overbrace{\sum_{j=0}^{i-1} n_j}^{\text{previous clusters}} - \underbrace{n_c + i + 1}_{\text{remaining clusters}}\right]$$

**end for**
$n_{n_c-1} \leftarrow n_{\max} - \sum_{i=0}^{n_c-2} n_i$       ▷ set the number of objects in the final cluster

---

**# select images**
**for** $i$ in range$(0, n_c)$ **do**
     $c_i \leftarrow \text{random\_cluster}(\mathcal{C} \setminus \bigcup_{j=0}^{i-1} \{c_j\})$       ▷ select a random, unique cluster as the $i^{\text{th}}$ cluster
     $\mathcal{I}_i \leftarrow \text{select}(n_i, c_i, \mathcal{O})$       ▷ select $n_i$ random images in cluster $c_i$ from $\mathcal{O}$
**end for**
$\mathcal{I} \leftarrow \bigcup_{i=0}^{n_c-1} \mathcal{I}_i$
$\tilde{\boldsymbol{I}} \leftarrow \text{select}(1, \mathcal{B})$       ▷ select a random background image from $\mathcal{B}$

---

**# compose the image and pseudo ground-truth**
$\mathbf{B} \leftarrow [\,]$       ▷ initialise an empty list for the object bounding boxes
**for** $\boldsymbol{I}$ in $\mathcal{I}$ **do**       ▷ iterate over all object images
     $s \leftarrow \text{get\_random\_size}(\boldsymbol{I}, \mathcal{I})$       ▷ get a random size, which is correlated for all images in $\mathcal{I}$
     $p \leftarrow \text{get\_random\_position}(s)$       ▷ get a random position for the current image
     $\boldsymbol{I}_r \leftarrow \text{resize}(\boldsymbol{I}, s)$       ▷ resize $\boldsymbol{I}$, if using segmentations, this involves cutting the object
     $\tilde{\boldsymbol{I}} \leftarrow \text{paste}(\tilde{\boldsymbol{I}}, \boldsymbol{I}_r, p)$       ▷ paste $\boldsymbol{I}_r$ into $\tilde{\boldsymbol{I}}$ at position $p$, if $\boldsymbol{I}$ is a target object, place it on top
     **if** $\boldsymbol{I} \in \mathcal{I}_0$ **then**
         $\mathbf{b} \leftarrow \text{box}(p, s)$       ▷ create the bounding box of the current object if $\boldsymbol{I}$ is a target object
         $\mathbf{B}.\text{append}(\mathbf{b})$       ▷ append the bounding box to the list of all object boxes
     **end if**
**end for**
$\mathcal{S} \leftarrow \text{crop\_exemplars}(\tilde{\boldsymbol{I}}, \mathbf{B}, E)$       ▷ create $E$ exemplar crops
$\mathbf{y} \leftarrow \text{create\_density\_map}(\mathbf{B})$       ▷ create the density map
**return** $\tilde{\boldsymbol{I}}, \mathcal{S}, \mathbf{y}$

---

### A.3 CONNECTED COMPONENTS BASELINE

We evaluate the connected components baseline on the FSC-147 training set to find the best values for its three thresholds $p_{\text{att}}$, $n_{\text{head}}$, and $p_{\text{size}}$. To this end, we run an exhaustive grid search by testing all combinations of the following settings:

$$p_{\text{att}} \in \{0.1, 0.2, ..., 0.9, 0.95, 0.99\}, \tag{4}$$
$$n_{\text{head}} \in \{1, ..., 12\}, \tag{5}$$
$$p_{\text{size}} \in \{0, 0.01, 0.02, 0.05, 0.1, 0.2\} \tag{6}$$

This results in 792 configurations, where we pick the setup with the lowest MAE on the whole FSC-147 training set which results in the following thresholds: $p_{\text{att}} = 0.7$, $n_{\text{head}} = 10$, and $p_{\text{size}} = 0$.

## A.4 INFERENCE DETAILS

We follow Liu et al. (2022) in our evaluation procedure. Specifically, we resize the inference image to a height of 384 keeping the aspect ratio fixed and scan the resulting image with a window of size $384 \times 384$ and a stride of 128 pixels. The density maps of the overlapping regions are averaged when aggregating the count of the entire image. If the image contains very small objects, defined as at least one exemplar with a width and height of less than 10 pixels, the image is divided into a $3 \times 3$ grid. Each of the 9 tiles is then resized to a height of 384 pixels and processed independently. The prediction for the original image is obtained by combining the individual predictions. Additionally, we apply the same test-time normalisation: We normalise the predicted count by the average sum of the density map areas that correspond to the exemplars if it exceeds a threshold of 1.8.

## A.5 SELF-SUPERVISED SEMANTIC COUNTING DETAILS

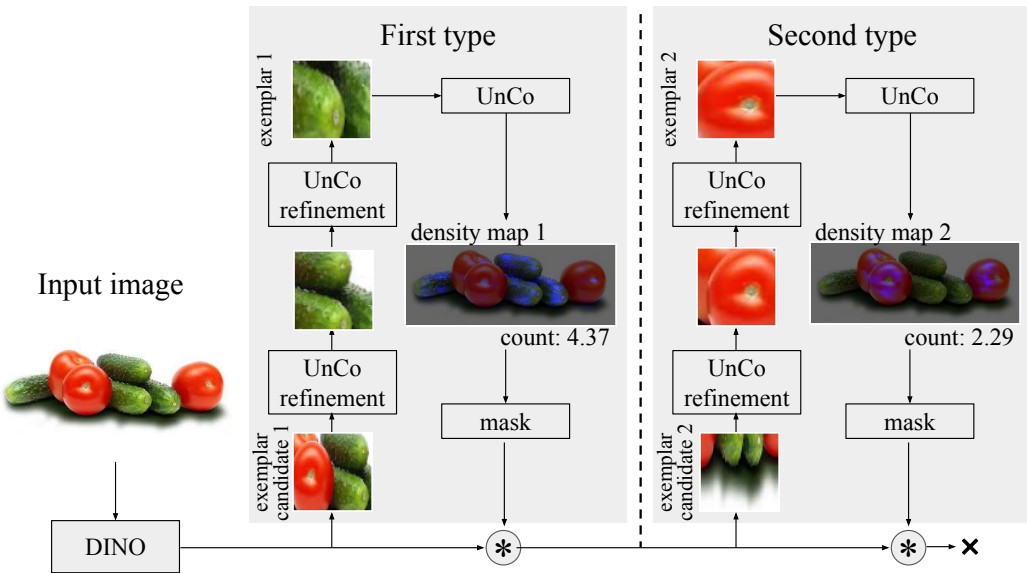

Figure 4: **Self-supervised semantic counting.** To predict the number of objects without any prior, the model uses its DINO backbone to get initial exemplar candidates, which it subsequently refines and uses to predict density maps for the discovered object types.

**Problem overview**  In the self-supervised semantic counting setup, the model automatically identifies the number of object types and appropriate exemplars to predict the density map for each category. Figure 4 illustrates this process.

**Processing the input image**  We first obtain the feature map of the input image $I$ resized to $384 \times 384$ pixels using the image encoder: $\mathbf{x} = \Phi(I) \in \mathbb{R}^{h \times w \times d}$. In addition, we extract the final CLS-attention map $\mathbf{a} \in \mathbb{R}^{h \times w}$ where we take the average across the attention heads and only consider the self-attention for the $h \cdot w$ patches ignoring the CLS-token itself.

**Proposing an exemplar candidate**  After processing the input image, the model determines an exemplar candidate to predict the density map $\hat{\mathbf{y}}^{(t)}$ of the first type $t = 1$. To achieve this, we blur $\mathbf{a}$ by applying a Gaussian kernel of size 3 with sigma 1.5 resulting in $\mathbf{a}^{(1)}$ and identify the position $(y_{\max}^{(1)}, x_{\max}^{(1)}) = \arg\max_{(y,x)} \mathbf{a}_{y,x}^{(1)}$ of the patch with the maximum attention. Subsequently, we compute a binary feature map $\mathbf{b}^{(1)} = \mathbf{a}^{(1)} > 0.5 \cdot \max \mathbf{a}^{(1)}$ and obtain the connected component which includes the position $(y_{\max}^{(1)}, x_{\max}^{(1)})$. The crop of $I$ corresponding to this component is taken as the exemplar candidate $E_{\text{candidate}}^{(1)}$.

**Refining the exemplar** Using $\boldsymbol{I}$ and $\boldsymbol{E}_{\text{candidate}}^{(1)}$, we predict a density map

$$\hat{\mathbf{y}}_{\text{candidate}}^{(1)} = f_\Theta(\boldsymbol{I}, \{\boldsymbol{E}_{\text{candidate}}^{(1)}\}), \tag{7}$$

where we can skip the image encoder $\Phi$ by reusing the feature map $\mathbf{x}$. $\hat{\mathbf{y}}_{\text{candidate}}^{(1)}$ is used to obtain a refined exemplar $\boldsymbol{E}_{\text{refined}}^{(1)}$. To this end, we binarise the density map $\hat{\mathbf{y}}_{\text{candidate}}^{(1)}$ by selecting values greater than 20% of its maximum and receive the second largest connected component, where we assume the largest component represents the background and the second largest corresponds to an object of the first type. We take a crop $\boldsymbol{E}_{\text{crop}}^{(1)}$ of $\boldsymbol{I}$ at the location of this component. Since $\boldsymbol{E}_{\text{crop}}^{(1)}$ might contain adjacent objects of different categories, we construct $\boldsymbol{E}_{\text{refined}}^{(1)}$ by taking another center crop of $\boldsymbol{E}_{\text{crop}}^{(1)}$ reducing each dimension by a factor of 0.3. The refinement step can be repeated multiple times by using $\boldsymbol{E}_{\text{refined}}^{(1)}$ as the new candidate in Equation (7). In practice, we use two refinement iterations.

**Predicting the count** We employ test-time normalisation and take the density map $\hat{\mathbf{y}}^{(1)}$ obtained using Equation (7) and replacing $\boldsymbol{E}_{\text{candidate}}^{(1)}$ with $\boldsymbol{E}_{\text{refined}}^{(1)}$ as final prediction for the first type.

**Counting multiple categories** To obtain predictions for more categories, we repeat these steps starting with a new maximum $(y_{\max}^{(2)}, x_{\max}^{(2)})$ after masking out the patches in $\mathbf{a}^{(1)}$ which correspond to the current category resulting in a new attention map $\mathbf{a}^{(2)}$. We identify these patches using two heuristics: First, we mask out patches where $\hat{\mathbf{y}}_{\text{candidate}}^{(1)}$ or $\hat{\mathbf{y}}_{\text{refined}}^{(1)}$, resized to match the dimensions of $\mathbf{a}^{(1)}$, predict a value higher than or equal to 0.5. Second, we set the attention to 0 in an area of $5 \times 5$ patches centred around $(y_{\max}^{(1)}, x_{\max}^{(1)})$ to prevent $(y_{\max}^{(2)}, x_{\max}^{(2)}) = (y_{\max}^{(1)}, x_{\max}^{(1)})$. Since every object should only be counted once and to facilitate the knowledge of previous iterations, we subtract the sum of the density maps of previous iterations, from the current prediction:

$$\hat{\mathbf{y}}^{(t)} = \max\left(f_\Theta(\boldsymbol{I}, \{\boldsymbol{E}_{\text{candidate}}^{(t)}\}) - \sum_{t'=1}^{t-1} \max\left(\hat{\mathbf{y}}^{(t')}, \mathbf{0}\right), \mathbf{0}\right) \tag{8}$$

This procedure keeps detecting exemplars and making predictions for new categories $t$ until the maximum remaining attention value is less than 20% of the original maximum at which point we assume that all salient object types have been detected.

**Evaluating the importance of the refinement steps** Especially the first refinement step is crucial to obtain meaningful exemplars as illustrated in Figure 4. While the initial candidates, which are only based on the DINO backbone, successfully highlight the salient objects, they fail to focus on a single object type. A single refinement step based on UnCo solves this issue which indicates the importance of UnCo's self-supervised training for this task. The second refinement step has a more subtle impact on the exemplar quality by reducing the number of objects in each exemplar. Based on these self-supervised exemplars, UnCo produces counts close to the true number of objects.

**Limitation** While these results are promising, they are only a qualitative exploration and are intended to highlight a potential avenue for future work. The creation of an evaluation dataset for the semantic counting task as well as the development of a metric to measure exemplar quality are required for a more thorough evaluation of this use case.

# B  MORE DETAILS OF SELF-COLLAGES

## B.1  UNDERLYING ASSUMPTIONS FOR SELF-COLLAGES

In Section 4.1, we describe the construction of Self-Collages with ImageNet-1k and SUN397 dataset. Our underlying assumptions are twofold:

1. Images in the SUN397 dataset do not contain objects to serve as the background for our Self-Collages.
2. Images in the ImageNet-1k dataset feature a single salient object to obtain correct pseudo labels.

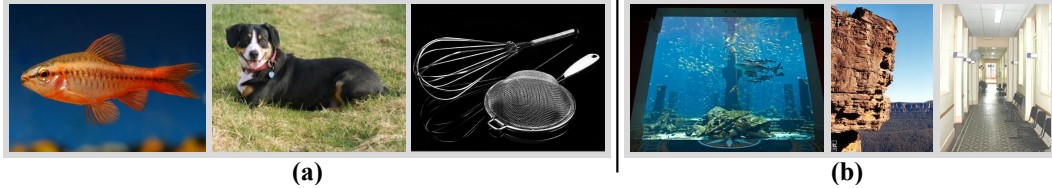

**(a)**          **(b)**

Figure 5: **ImageNet-1k and SUN397 images.** **(a)** Example images from the ImageNet-1k dataset (Russakovsky et al., 2015). **(b)** Example images from the SUN397 dataset (Xiao et al., 2010). While the figures in the SUN397 dataset may contain multiple objects, there is no clearly salient object.

Figure 5 shows three samples from ImageNet-1k and SUN397. We can see that even though images in the latter do contain objects, see *e.g.* the fish in the aquarium, they are usually not salient. Hence, the first assumption is still reasonable for constructing Self-Collages. We acknowledge that this assumption has its limitations and introduces noises for Self-Collages, *e.g.* salient objects exist in some images from SUN397. In Appendix C.3, we consider variants of the default setup to investigate the robustness of our method against violations of this assumption.

The second assumption is crucial to derive a strong supervision signal from unlabelled images. While some ImageNet-1k images contain multiple salient objects, see *e.g.* the right-most image in Subfigure **a**, the final performance of our method shows that the model is able to learn the task even with this noisy supervision. Some techniques such as filtering images based on their segmentation masks, might be able to further improve the supervision signal. We consider these methods as future works.

## B.2 EXAMPLES OF SELF-COLLAGES

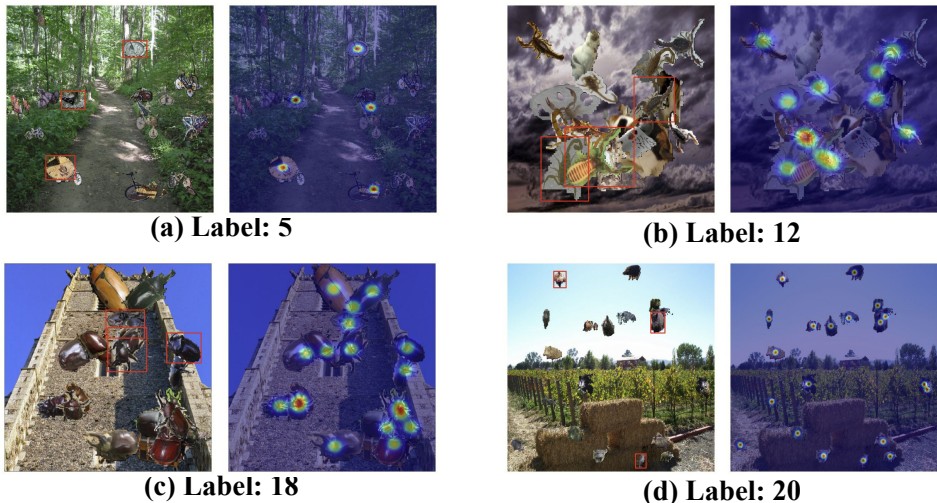

**(a) Label: 5**          **(b) Label: 12**

**(c) Label: 18**          **(d) Label: 20**

Figure 6: **Example Self-Collages.** The Subfigures show Self-Collages for different counts. The red boxes indicate the exemplars and the heatmaps show the pseudo ground-truth density maps.

Figure 6 shows different examples of Self-Collages. While the whole pipeline does not rely on any human supervision, the Self-Collages contain diverse types that align with human concepts from objects like bicyles (Subfigure **a**) to animals such as beetles (Subfigure **c**). Likewise, the unsupervised segmentation method proposed by Shin et al. (2022) successfully creates masks for the different instances. The correlated sizes lead to similarly sized objects in each Self-Collage with some samples containing primarily small instances (Subfigures **a** and **d**) and others having mainly bigger objects (Subfigures **b** and **c**). Adding to the diversity of the constructed samples, some Self-Collages show significantly overlapping objects (Subfigure **b**) while others have clearly

separated entities (Subfigure **d**). Since the total number of pasted objects is constant, the amount of pasting artefacts in each image does not provide any information about the target count. The density maps indicate the position and number of target objects, overlapping instances result in peaks of higher magnitude (Subfigure **b** and **c**). By computing the parameters of the Gaussian filter used to construct the density maps based on the average bounding box size, the area covered by density mass correlates with the object size in the image.

### B.3 COLLAGES IN OTHER WORKS

The idea of deriving a supervision signal from unlabelled images by artificially adding objects to background images, the underlying idea behind Self-Collages, is also used in other domains such as object discovery and segmentation.

Arandjelović & Zisserman (2019) propose a generative adversarial network, called copy-pasting GAN, to solve this task in an unsupervised manner. They train a generator to predict object segmentations by using these masks to copy objects into background images. The learning signal is obtained by jointly training a discriminator to differentiate between fake, *i.e.* images containing copied objects, and real images. A generator that produces better masks results in more realistic fake images and has therefore higher chances of fooling the discriminator. During inference, the generator can be used to predict instance masks in images. By contrast, our composer module $g$ is only used during training to construct samples. Hence, we do not need an adversarial setup and, without the need to update $g$ during training, the composer module does not have to be differentiable.

To boost the performance of instance segmentation methods, Zhao et al. (2022) propose the X-Paste framework. It leverages recent multi-modal models to either generate images for different object classes from scratch or filter web-crawled images. After creating instance masks and filtering these objects, the resulting images are pasted into background images. Generated samples can then be used in isolation or combined with annotated samples to train instance segmentation methods. Compared to this work, our goal is not to train instance segmentation but counting methods where the correct number of objects in the training samples is more important than the quality of the individual segmentations. Partly due to this, our method is conceptually simpler than X-Paste and does not require, for example, multiple filtering steps.

## C FURTHER RESULTS

In this section, we further analyse UnCo's performance in out-of-domain settings (see Appendix C.1) and explore ways to improve UnCo's default setup based on recent advances in self-supervised representation learning (see Appendix C.2). Finally, we evaluate our method under different data distributions in Appendix C.3.

### C.1 GENERALISATION TO OUT-OF-DOMAIN COUNT DISTRIBUTIONS

In this section, we examine UnCo's performance on the different FSC-147 subsets, as presented in Table 1, to investigate its generalisation capabilities to new count distributions. We consider the results on the three subsets *low*, *medium*, and *high* with an average of 12, 27, and 117 objects per image respectively. Since UnCo is trained with Self-Collages of 11 target objects on average, the count distribution of FSC-147 *low* can be seen as in-distribution while the *medium* and *high* subsets are increasingly more out-of-distribution (OOD).

Unsurprisingly, the model performs worse on subsets whose count distributions deviate more from the training set. Looking at FSC-147 *medium*, the RMSE changes only slightly considering the significant increase in the number of objects compared to *low*. When moving to the *high* subset whose count distribution differs significantly from the training set, the error increases substantially.

These trends can also be seen in Figure 7. Figure 7a shows that the model's predictions are distributed around the ground-truth for FSC-147 *low* and *medium*. On the *high* subset, the model tends to underestimate the number of objects in the images. The relative error becomes increasingly negative as these counts increase (see Figure 7b) illustrating the limitations of generalising to OOD count ranges.

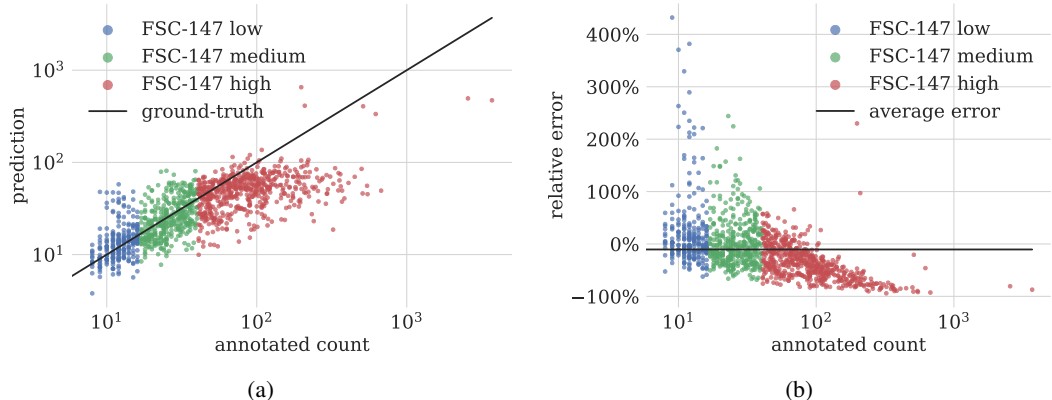

(a)                                                      (b)

Figure 7: **UnCo's predictions.** Figure 7a compares the model's predictions on the different FSC-147 test subsets with the ground-truth. Figure 7b visualises the relative error for these predictions.

Table 8: **Improving UnCo's performance.** We explore several variations of UnCo's default setup, highlighted in grey , to further close the performance gap to its supervised counterparts.

| | | | FSC-147 | | | FSC-147 **low** | | |
|---|---|---|---|---|---|---|---|---|
| Backbone | similarity | refinement | MAE↓ | RMSE↓ | $\tau$ ↑ | MAE↓ | RMSE↓ | $\tau$ ↑ |
| DINOv2 | ✗ | ✗ | 34.35 | 132.02 | 0.63 | 4.10 | 7.34 | 0.32 |
| DINOv2 | ✓ | ✗ | 31.06 | 119.35 | 0.67 | 3.98 | 7.58 | **0.33** |
| DINOv2 | ✓ | ✓ | **28.67** | **118.40** | **0.71** | **3.88** | **7.03** | 0.33 |
| DINO | ✗ | ✗ | 35.77 | 130.34 | 0.57 | 5.60 | 10.13 | 0.27 |

While this is expected, we can assume that a model that learned a robust notion of numerosity gives higher count estimates for images containing more objects even in these OOD settings. This can be quantified using the rank correlation coefficient Kendall's $\tau$. Interestingly, the correlation coefficient on FSC-147 *low* and *high* is almost the same, the highest value for $\tau$ is achieved on the *medium* subset (see Table 1). This suggests that while UnCo's count predictions become less accurate for out-of-distribution samples, the model's concept of numerosity as learned from Self-Collages generalises well to much higher count ranges.

## C.2 IMPROVING THE DEFAULT SETUP

Having established the default UnCo setup and compared it to several methods, we now explore ways to further improve UnCo's performance based on the very recent DINOv2 (Oquab et al., 2023) backbone. Building on this change, we investigate two further modifications: exploiting cluster similarity and refining the model's predictions. Table 8 shows the results for the different variations.

## C.2.1 UPDATING THE BACKBONE

First, we update the DINO backbone (Caron et al., 2021) with the newer DINOv2 (Oquab et al., 2023). More specifically, we employ the ViT-B model with a patch size of 14. Due to the different patch size, we change the resolution of the exemplars slightly from $64 \times 64$ to $70 \times 70$. In addition, we update the sliding window's dimension to $392 \times 392$ during inference.

The results in Table 8 show that updating the backbone improves the overall performance on the whole FSC-147 dataset, with a small increase in RMSE by 1.3% being the only exception. In particular, this change seems to be beneficial for images with lower counts, the corresponding metrics improve by 19-28%. This demonstrates UnCo's ability to take advantage of recent advances in self-supervised representation learning as discussed in Section 4.3.

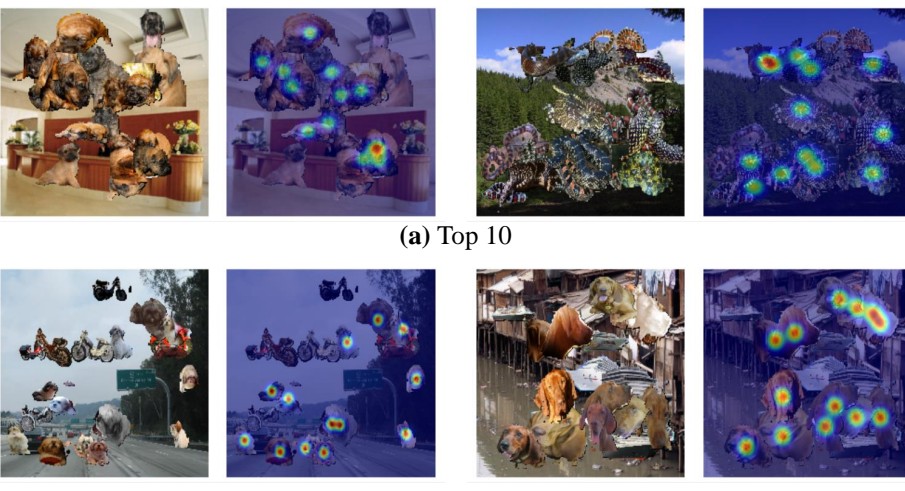

**(a)** Top 10

**(b)** Bottom 1000

Figure 8: **Constructing Self-Collages based on cluster similarity.** The similarity between clusters can be used when constructing Self-Collages, the pseudo ground-truth on the right of each image indicates the objects of the target cluster. Subfigure **a** shows Self-Collages where the non-target cluster was randomly chosen among the 10 clusters most similar to the target cluster. Due to the high similarity, objects in the target and non-target clusters are not easily distinguishable. By contrast, the examples in Subfigure **b** use the 1000 most different clusters. The two types in each image are visually very distinct.

### C.2.2 CLUSTER SIMILARITY

Since multiple object types are pasted to generate our training images, we next exploit information about their similarity. We hypothesise that increasing the difficulty of the counting task during training by selecting non-target clusters that are similar to the target clusters could facilitate the learning process. This idea resembles the use of hard negatives in contrastive learning which has been shown to improve the robustness of learned representations (Xuan et al., 2020; Ge et al., 2021; Zhang et al., 2022). Unlike in supervised setups with manually annotated classes, information about the similarity of object clusters is readily available in our self-supervised setup and can be computed simply as the negative Euclidean distance between the cluster centres. Figure 8 shows the effect of cluster similarity on the constructed Self-Collages.

In Table 9, we compare different ways of exploiting the similarity information. It can be seen that selecting non-target clusters that are similar to the target cluster significantly improves the performance. Importantly, picking non-target clusters that are too similar to the target cluster severely harms the training process. Considering the qualitative samples in Figure 8, very similar object types are almost visually indistinguishable even for humans. Hence, we use the 100 clusters most similar to the target cluster excluding the top 10. Unlike the updated backbone, this change improves the performance especially on images with higher counts as seen in Table 8.

### C.2.3 PREDICTION REFINEMENT

Lastly, we modify our evaluation protocol to mitigate the count distribution shift between training and the FSC-147 test set as discussed in Appendix C.1. To this end, we employ a refinement strategy which aims in particular at improving the predictions for images with high object counts. First, we obtain a prediction using the default inference setup described in Appendix A.4. Then, if the model predicts more than 50 objects, we utilise the same setup as employed for small objects where we split the image into 9 tiles and predict the counts for each of them independently before aggregating the final prediction (see Appendix A.4).

Table 8 shows that employing this evaluation protocol further improves the performance resulting in an MAE of 28.67 on FSC-147. Combining all three modifications reduces the MAE compared

Table 9: **Using cluster similarities to construct Self-Collages.** We use cluster similarities in the composer module $g$ to select non-target clusters during Self-Collage construction. Top $X$ describes the setup where we pick the non-target cluster among the $X$ clusters that are most similar to the target cluster. Likewise, the clusters are chosen from the set of the $X$ most dissimilar clusters in the bottom $X$ setting. If we specify a range $X$-$Y$, the set of possible non-target clusters is equal to top $Y$ without the elements in top $X$. All setups use DINOv2 as backbone.

| | FSC-147 | | |
|---|---|---|---|
| similarity range | MAE↓ | RMSE↓ | $\tau$ ↑ |
| ✗ | 34.35 | 132.02 | 0.63 |
| top 10 | 36.67 | 130.30 | 0.59 |
| 10-100 | **31.06** | **119.35** | **0.67** |
| bottom 1000 | 34.71 | 131.12 | 0.63 |

to the default UnCo setup by 20% on FSC-147 and 31% on FSC-147 *low* which narrows the gap to the supervised counterparts and highlights the potential of unsupervised counting based on Self-Collages.

### C.3 SELF-COLLAGES BASED ON DIFFERENT DATASETS

In Table 10, we ablate the datasets used for object ($\mathcal{O}$) and background images ($\mathcal{B}$) to investigate the behaviour of our method under different data distributions. Figure 9 shows training samples for the ablated setups. We first describe the different datasets in Appendix C.3.1, followed by the ablation results in Appendix C.3.2.

#### C.3.1 DATASET ABLATIONS

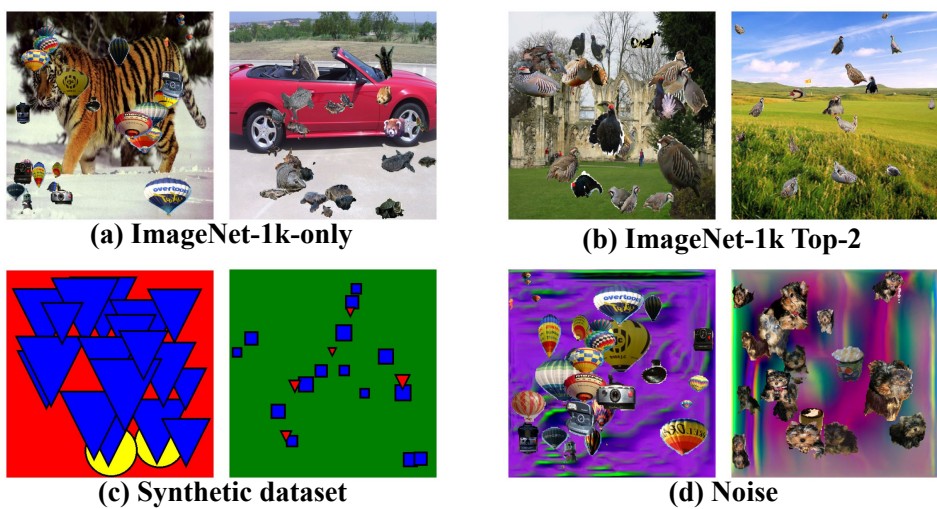

**(a) ImageNet-1k-only**   **(b) ImageNet-1k Top-2**

**(c) Synthetic dataset**   **(d) Noise**

Figure 9: **Dataset ablations.** We modify the training dataset by using different datasets for $\mathcal{O}$ and $\mathcal{B}$ to investigate the effect on the final counting performance.

**ImageNet-1k-only**   To simplify the construction process, we experiment with using the ImageNet-1k dataset for both, $\mathcal{O}$ and $\mathcal{B}$. To make sure that the counting task is not affected by the background, we exclude all images in the two clusters used for the object images before randomly selecting the background image.

**ImageNet-1k Top-2**   As a simpler version of the default setup, we filter the ImageNet-1k dataset to only use the images in the two biggest clusters, which we call ImageNet-1k Top-2. Looking at

Table 10: **Using different datasets to construct Self-Collages.** We vary the datasets used by the composer module $g$ to obtain object ($\mathcal{O}$) and background images ($\mathcal{B}$) when constructing Self-Collages. The default setting is highlighted in grey .

| $\mathcal{O}$ | $\mathcal{B}$ | FSC-147 | | | FSC-147 **low** | | |
|---|---|---|---|---|---|---|---|
| | | MAE↓ | RMSE↓ | $\tau \uparrow$ | MAE↓ | RMSE↓ | $\tau \uparrow$ |
| ImageNet-1k | | 35.68 | 130.10 | 0.56 | 6.78 | 12.43 | 0.25 |
| ImageNet-1k Top-2 | SUN397 | 39.47 | 132.37 | 0.41 | 12.08 | 17.84 | 0.16 |
| Synthetic dataset | | 45.16 | 144.60 | 0.26 | 7.03 | 11.04 | 0.23 |
| ImageNet-1k | Noise | **34.43** | **128.73** | **0.60** | 5.94 | 10.72 | 0.26 |
| ImageNet-1k | SUN397 | 35.77 | 130.34 | 0.57 | **5.60** | **10.13** | **0.27** |

Subfigure **b** in Figure 9, these clusters seem to correspond to two different types of birds. This reduces the number of images from 1,281,167 to 1,266. Because this subset only features two clusters, every Self-Collage contains the same object types.

**Synthetic dataset**   We construct a synthetic dataset based on simple shapes. To this end, we combine three types of shapes, `squares`, `circles`, and `triangles`, and four different colours, `red`, `green`, `blue`, and `yellow`, to obtain a total of 12 possible object types. After randomly picking two different types, we select a random background colour amongst the colours not used for the objects.

**Noise**   We use the StyleGAN-Oriented dataset proposed by Baradad Jurjo et al. (2021), which is based on a randomly initialised StyleGANv2 (Karras et al., 2020), as noise dataset to draw background images from. In total, it contains 1.3M synthetic images. We refer to the original work (Baradad Jurjo et al., 2021) for more details.

### C.3.2   DATASET ABLATION RESULTS

**Image diversity improves generalisability**   It can be seen, that gradually decreasing the image diversity, by using the same dataset for $\mathcal{O}$ and $\mathcal{B}$, using only a very small ImageNet-1k subset for $\mathcal{O}$, or using a fully synthetic dataset, harms the performance on the FSC-147 dataset.

**Simple objects are sufficient for low counts only**   While using the fully synthetic dataset yields the worst results on FSC-147, the performance on the FSC-147 *low* subset is comparable to the setup using only ImageNet-1k. This indicates the importance of more realistic objects for the generalisability to higher counts. At the same time, synthetic objects seem to be sufficient for learning to predict the number of objects in images with few instances.

**Synthetic but diverse backgrounds perform similarly to real images**   When replacing SUN397 with a noise dataset, the performance on the whole FSC-147 dataset improves while being slightly worse on the FSC-147 *low* subset.

**Self-Collages are robust against violations of the background assumption**   Comparing the setup which uses the ImageNet-1k dataset for $\mathcal{O}$ and $\mathcal{B}$ to the default setting, we can see that even by explicitly violating the assumption that there are no salient objects in $\mathcal{B}$, the performance is not significantly affected. However, by using a noise dataset as $\mathcal{B}$ where the assumption holds, we can further improve the performance on FSC-147. This indicates that while our method is robust to violations of the aforementioned assumption, using artificial datasets where the assumption is true, can be beneficial.

## C.4 COMPARING FASTERRCNN AND UNCO

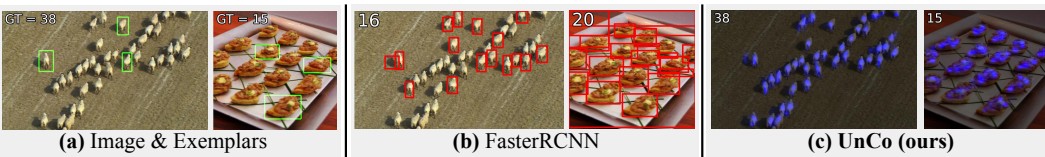

| **(a)** Image & Exemplars | **(b)** FasterRCNN | **(c)** UnCo (ours) |

Figure 10: **When UnCo works better than FasterRCNN.** Given two images with exemplars **(a)** we compare the output predictions of FasterRCNN **(b)** and of our model **(c)**. We find that FasterRCNN either misses most objects in high-density settings or detects non-target instances which is because the model cannot utilise any prior knowledge in the form of exemplars.

## C.5 QUALITATIVE RESULTS

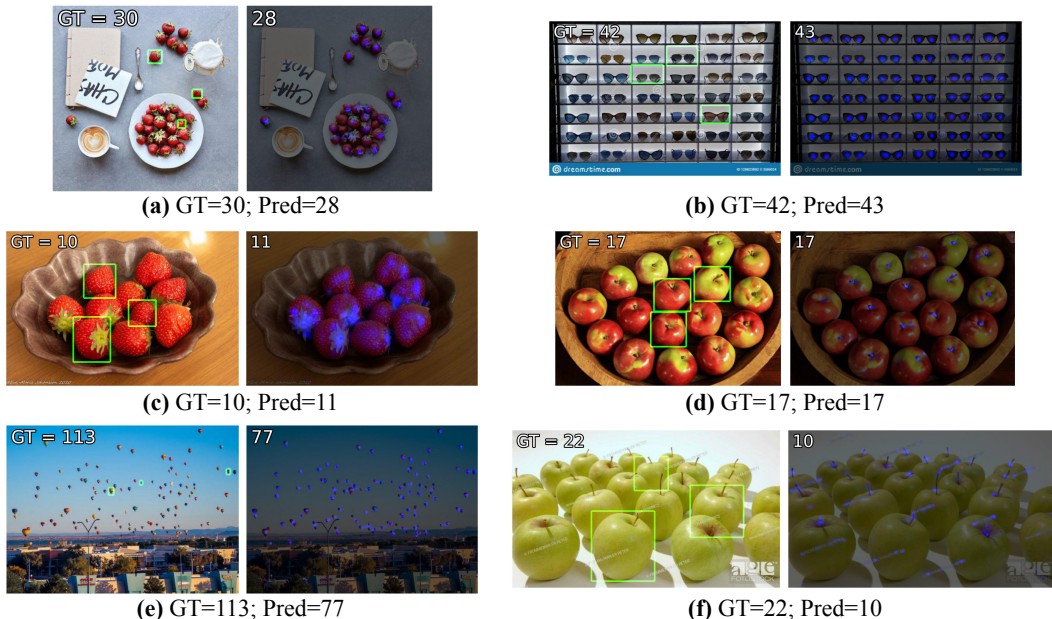

**(a)** GT=30; Pred=28

**(b)** GT=42; Pred=43

**(c)** GT=10; Pred=11

**(d)** GT=17; Pred=17

**(e)** GT=113; Pred=77

**(f)** GT=22; Pred=10

Figure 11: **Qualitative UnCo results.** We show UnCo's predictions on six different images from the FSC-147 test set. The green boxes represent the exemplars and the number in the top-left of each image indicate the ground-truth and predicted count for each sample. Our prediction is the sum of the heatmap rounded to the nearest integer.

Figure 11 visualises predictions of our model UnCo on the FSC-147 dataset. The model predicts good count estimates even for images with more than twice as many objects as the maximum number of objects seen during training (Subfigure **b**). In general, the model successfully identifies the object type of interest and focuses on the corresponding instances even if they only make up a small part of the entire image (Subfigure **a**). However, UnCo still misses some instances in these settings.

For very high counts (Subfigure **e**) and images with artefacts such as watermarks (Subfigure **f**), the model sometimes fails to predict a count close to the ground-truth.

