# OpenReview forum: "Learning to Count without Annotations"
_ICLR.cc/2024/Conference — ICLR 2024 Conference Withdrawn Submission_

### Official Review · Reviewer_1EYE · 2023-10-31

**Soundness:** 3 good
**Presentation:** 2 fair
**Contribution:** 1 poor
**Rating:** 3
**Confidence:** 5

**Summary:**

This manuscript targets on few shot counting model training without annotations. Specifically, the authors build a synthetic dataset to provide supervision signal to the object counter, and utilize a DINO and Vision Transformer based architecture to make prediction of density map.

**Strengths:**

1. This manuscript is sound in making adequate explaination to the results and experimental analysis;
2. The writing quality of this manuscript is ok to make me get the points.

**Weaknesses:**

1. The motivation of this work is poor. I am still confused on why we should build such a sythetic dataset from others to get some supervision signal to train a unsupervised counter.
2. From the data perspective, these generated data are without double to be filled with artefact and the solution in this manuscript is just the copy-paste, whose contribution is limited.
3. The counting model utilized in this manuscript is not totally original, which seems to be the DINO + ViT.
4. It is evident that the authors omitted some methods in unsupervised counter (Completely Self-Supervised Crowd Counting via Distribution Matching-ECCV22), or foundation model based method (Can SAM Count Anything? An Empirical Study on SAM Counting) & (Training-free Object Counting with Prompts).

**Questions:**

None

---

### Official Review · Reviewer_21rs · 2023-10-31

**Soundness:** 3 good
**Presentation:** 3 good
**Contribution:** 2 fair
**Rating:** 5
**Confidence:** 5

**Summary:**

This paper focuses on the unsupervised object-counting task that does not require any manual annotations. To this end, the authors construct “SelfCollages”, images with various pasted objects as training samples, that provide a rich learning signal covering arbitrary object types and counts. Experiments on the counting dataset demonstrate the effectiveness of the proposed method.

**Strengths:**

The unsupervised counting task is a challenging task, and it is appealing to see the authors propose a practical way.

The proposed method even outperforms simple baselines and generic models such as FasterRCNN and DETR.

**Weaknesses:**

1. The experiments are not convincing. There are two pioneering works (CrowdCLIP[1] and CSCCNN ) that also focus on the unsupervised counting task. However, the authors do not discuss or compare with them. I would like to see a comprehensive comparison.
2. The evaluated FSC-147 dataset is not very challenging. I suggest the authors try to conduct experiments on the crowd datasets, which are usually dense and challenging. Compared with CrowdCLIP[1] and CSC-CCNN[2] will make the paper more solid.
3. It is better to add a subsection to discuss the weakly/semi-supervised counting methods that also reduce the annotation cost.
4. The motivation for pasting different images on top of a background is not clear.
5. Do the authors try to utilize other cluster algorithms unless the K-means？

[1] CrowdCLIP: Unsupervised Crowd Counting via Vision-Language Model. CVPR 2023.
[2] Completely self-supervised crowd counting via distribution matching. ECCV 2022.

**Questions:**

see weakness

---

### Official Review · Reviewer_HmkX · 2023-11-02

**Soundness:** 3 good
**Presentation:** 3 good
**Contribution:** 1 poor
**Rating:** 3
**Confidence:** 4

**Summary:**

This paper introduces a method for counting objects without annotations. It leverages DINO and N-cut to extract object patches and then randomly places them into a background image, allowing for the acquisition of localization labels without annotation. Additionally, it trains a counting model based on CounTR but with the DINO backbone to count objects.

**Strengths:**

1. The authors propose a method to generate synthetic data for object counting and implement an unsupervised object counting approach.
2. They utilize the DINO backbone to create a counting model similar to CounTR.

**Weaknesses:**

1. The approach of creating synthetic data by copying segmentation results from one image to another is a well-known technique in segmentation [1]. However, this paper applies it to object counting.
2. The trained model's performance is not satisfactory, particularly in FSC-147 high, which is the primary objective of counting dense and small objects.
3. The motivation for the counting task is to abstract information from dense scenes that detection models struggle with, particularly partial and occluded objects. However, the proposed method does not effectively handle partial or occluded objects, which contradicts the motivation of the counting task.

[1] Ghiasi, Golnaz, et al. "Simple copy-paste is a strong data augmentation method for instance segmentation." CVPR, 2021.

**Questions:**

1. How does the model perform if fine-tuned on FSC-147?
2. How does the model perform on a specific dataset without retraining? For instance, previous methods have conducted adaptation on the CARPK dataset.
3. Why is $n_{max}$ set as a very small value ($n_{max} = 20$)? A higher count might be more suitable for a counting model since powerful detection models can handle it in sparse scenes.
4. Although SC-147 is split into low/medium/high in experiments, the overall performance should also be reported in the corresponding tables and sections.
5. Table 5 seems unfair. $n_{max}$ in UnCo is 20, making the trained model more suited to FSC-147 low (8-16 objects). CounTR is trained on the entire FSC-147 dataset, with counts ranging from 7 to 3731. The domains are different, and for a fair comparison, CounTR should be trained using only samples from FSC-147-low.
6. The average baseline in Table 1 seems unfair, particularly for FSC-147 low. Specifically, if the estimation density is 0, the MAE is the average GT count in FSC-147 low (8-16), which is much smaller than 37.

---

### Author Response · Authors · 2023-11-15
**Appreciation of Reviews and Withdrawal Decision**

We appreciate your review of our paper. After careful consideration, we have decided to incorporate your feedback to further improve our paper and withdraw our submission. Thank you for your comments.